# Role of Neuron–Glia Signaling in Regulation of Retinal Vascular Tone in Rats

**DOI:** 10.3390/ijms20081952

**Published:** 2019-04-20

**Authors:** Eriko Someya, Mari Akagawa, Asami Mori, Akane Morita, Natsuko Yui, Daiki Asano, Kenji Sakamoto, Tsutomu Nakahara

**Affiliations:** Department of Molecular Pharmacology, Kitasato University School of Pharmaceutical Sciences, 5-9-1 Shirokane, Minato-ku, Tokyo 108-8641, Japan; ml18111@st.kitasato-u.ac.jp (E.S.); pp14070@st.kitasato-u.ac.jp (M.A.); moria@pharm.kitasato-u.ac.jp (A.M.); moritaa@pharm.kitasato-u.ac.jp (A.M.); ps09256@st.kitasato-u.ac.jp (N.Y.); asanod@pharm.kitasato-u.ac.jp (D.A.); sakamotok@pharm.kitasato-u.ac.jp (K.S.)

**Keywords:** glial cell, neuronal cell, nitric oxide, neuronal nitric oxide synthase, retina

## Abstract

The interactions between neuronal, glial, and vascular cells play a key role in regulating blood flow in the retina. In the present study, we examined the role of the interactions between neuronal and glial cells in regulating the retinal vascular tone in rats upon stimulation of retinal neuronal cells by intravitreal injection of *N*-methyl-d-aspartic acid (NMDA). The retinal vascular response was assessed by measuring the diameter of the retinal arterioles in the in vivo fundus images. Intravitreal injection of NMDA produced retinal vasodilation that was significantly diminished following the pharmacological inhibition of nitric oxide (NO) synthase (nNOS), loss of inner retinal neurons, or intravitreal injection of glial toxins. Immunohistochemistry revealed the expression of nNOS in ganglion and calretinin-positive amacrine cells. Moreover, glial toxins significantly prevented the retinal vasodilator response induced by intravitreal injection of NOR3, an NO donor. Mechanistic analysis revealed that NO enhanced the production of vasodilatory prostanoids and epoxyeicosatrienoic acids in glial cells in a ryanodine receptor type 1-dependent manner, subsequently inducing the retinal vasodilator response. These results suggest that the NO released from stimulated neuronal cells acts as a key messenger in neuron–glia signaling, thereby causing neuronal activity-dependent and glial cell-mediated vasodilation in the retina.

## 1. Introduction

Neurovascular coupling refers to the relationship between neuronal activity and local blood flow in the retina and brain [1]. This response provides additional oxygen and nutrients to match the increased metabolic demands of active neurons. This homeostatic response was initially believed to be mediated by signals transmitted from neuronal cells to vascular cells; however, an increasing amount of evidence indicates that glial cells, such as astrocytes and Müller cells, also contribute to neurovascular coupling by interacting with neuronal and vascular cells [2,3,4].

Findings from previous studies indicate that retinal stimulation by flickering light (increasing neuronal activity) induces the dilation of retinal blood vessels and a consequent increase in the retinal blood flow. Interestingly, these responses are attenuated by the inhibition of neuronal nitric oxide (NO) synthase (nNOS) [5]. Moreover, glial toxins partially prevent the flickering light-stimulated increase in the retinal blood flow [3]. Recently, we found that glial toxins could attenuate the dilation of retinal arterioles induced by intravitreally injected NOR3, an NO donor [6]. These findings suggest that the nNOS-derived NO and the glial cells act as important contributors to the neuronal activity-dependent vasodilation in the retina. However, how NO modulates glial cell function and what factors released from glial cells are responsible for the vasodilator response remain to be determined.

Glutamate is a major excitatory neurotransmitter in the retina as well as in the brain. Synaptically released glutamate acts on *N*-methyl-d-aspartic acid (NMDA) receptors expressed either on retinal ganglion cells (RGCs), amacrine cells, or both, causing an increase in the intracellular Ca^2+^ levels. Furthermore, immunohistochemical studies have revealed the expression of nNOS in the inner plexiform layer (IPL), RGCs, and amacrine cells [6,7,8]. The increased intracellular Ca^2+^ concentration enhances the nNOS activity via calmodulin binding. Therefore, we hypothesized that glutamate-mediated activation of NMDA receptors on RGCs and amacrine cells facilitates nNOS-derived NO production in these cells. The increased NO levels, in turn, modulate the activity of glial cells to produce metabolites that induce vasodilator response in the retina.

In the present study, we investigated: (1) the effects of intravitreal injection of NMDA on the diameter of retinal arterioles; (2) the effects of glial toxins (disialoganglioside GD1b, l-alpha-aminoadipic acid, and fluorocitrate) on NMDA or NOR3-induced dilation of retinal arterioles; (3) what factors released from glial cells affect the vascular tone; and (4) what mechanisms are involved in the release of glial cell-derived vasoactive substances in rats.

## 2. Results

The baseline parameters were not different between the experimental groups (Table 1 and Table 2). 

Figure 1A shows representative fundus images before, and 10 and 30 min after the intravitreal injection of saline or NMDA (6 nmol). We observed a gradual increase in the diameter of retinal arterioles that reached its plateau level 5 to 10 min after the intravitreal injection of NMDA (Figure 1Ba). The increase in retinal arteriolar diameter persisted for at least 1 h. Intravitreal injection of NMDA had no significant effect on the mean arterial pressure and heart rate (Figure 1Bb,c).

To examine the possible involvement of NO in NMDA-induced vasodilation in the retina, we examined the effects of *N*^ω^-nitro-l-arginine methyl ester (l-NAME; a non-selective NOS inhibitor) and *N*^ω^-propyl-l-arginine (l-NPA; a selective nNOS inhibitor) on NMDA-induced response. We found that the NMDA-induced dilation of retinal arterioles was markedly attenuated in l-NAME- and l-NPA-treated rats (Figure 2A). Indomethacin (a cyclooxygenase inhibitor) also reduced the NMDA-induced vasodilation, and a further reduction in NMDA-induced response was observed with the combination of indomethacin and l-NPA (Figure 2B).

Cross-sections stained with anti-nNOS antibodies revealed nNOS immunoreactivities in the fibers in the IPL and nNOS-positive cells with strong and weak immunoreactivities in the ganglion cell layer (GCL) and inner nuclear layer (INL) (Figure 3A). Co-labeling of nNOS with βIII-tubulin indicated certain RGCs to express nNOS (Figure 3B). Consistent with previous observations [8], some calretinin-expressing amacrine cells in the INL expressed nNOS (Figure 3C). On the other hand, nNOS immunoreactivities were not detected in the calbindin-, parvalbumin-, tyrosine hydroxylase-, glycine transporter 1-positive cells (Appendix A).

We next examined the role of neuronal cells in NMDA-induced response in the retina using the retinal neuronal cell loss model. The nNOS-positive area in the IPL and the number of nNOS-expressing cells in the retina decreased in the NMDA (200 nmol)-induced neuronal cell loss model, whereas nNOS immunoreactivities in the choroid remained unaltered (Figure 4A,B). Quantitative analyses indicated that the nNOS-positive area in the IPL, the number of nNOS expressing cells in the GCL, and the number of strong and weak NOS-positive cells in the INL were significantly reduced in the neuronal cell loss model (Figure 4C–E). Thus, we found that nNOS-expressing neurons are highly susceptible to NMDA-induced retinal injury. Compared with controls, NMDA-induced retinal vasodilation was markedly reduced in the neuronal cell loss model (Figure 4F). Our previous studies demonstrated that the responsiveness of retinal arterioles to NO remained unaltered in the neuronal cell loss model [6]. Therefore, the damage to retinal vascular cells could not explain the reduced NMDA-induced retinal vasodilation.

Using glial toxins, we examined the contribution of glial cells to NMDA-induced vasodilation in the retina. The NMDA-induced retinal vasodilation was markedly attenuated in the presence of disialoganglioside-GD1b, a glial toxin (Figure 5A). The number of nNOS-expressing neurons and the network of glial fibrillary acidic protein (GFAP)-positive astrocytes were unaffected by intravitreal injection of disialoganglioside-GD1b. Therefore, we believe that dysfunction of glial cells reduced the NMDA-induced retinal vasodilation. No immunoreactivities of nNOS were detected in the GFAP-positive astrocytes (Figure 5B); however, several nNOS-positive cells were located near the GFAP-positive astrocytes in the GCL (Figure 5B), and nNOS-positive fibers were closely associated with the processes of GFAP-positive astrocytes in the IPL (Figure 5C).

An intravitreal injection of NOR3 (an NO donor, 5 nmol) increased the diameter of retinal arterioles, and the vasodilator responses were significantly prevented by disialoganglioside-GD1b (Figure 6A), l-alpha-aminoadipic acid (Figure 6B), and fluorocitrate (Figure 6C). These toxins did not affect the responsiveness of retinal blood vessels to NO [6]. Therefore, NOR3 injected into the vitreous cavity seems to preferentially act on the glial cells rather than on vascular cells and induce glial cell-dependent retinal vasodilator response.

Glial cells can release vasoactive metabolites of arachidonic acid, such as vasodilators, for example prostaglandin E_2_ (PGE_2_) and epoxyeicosatrienoic acids (EETs), and the vasoconstrictor 20-hydroxyeicosatetraenoic acid (20-HETE) [9,10]. We found that the cyclooxygenase inhibitor indomethacin (10 nmol), and the EET analog 14,15-epoxyeicosa-5(Z)-enoic acid (14,15-EE-5(Z)-E, 0.7 nmol) attenuated the retinal vasodilation induced by intravitreal injection of NOR3 (5 nmol). However, we did not observe any reduction in the NOR3-induced response by the inhibitor of ω-hydroxylase 17-octadecynoic acid (1.4 nmol) (Figure 7). These results suggest that NO stimulated the glial cells to release vasodilatory metabolites, i.e., prostanoids and EETs.

An increase in the intracellular Ca^2+^ concentrations activates phospholipase A_2_, which converts membrane phospholipids to arachidonic acid. Arachidonic acid, in turn, is subsequently metabolized to prostaglandins and EETs. In addition to Ca^2+^ influx from extracellular spaces, Ca^2+^ is mobilized within the cells from intracellular Ca^2+^ stores through two types of Ca^2+^ release channels: ryanodine receptors and inositol 1,4,5-trisphosphate (IP_3_) receptors. The ryanodine receptor antagonist dantrolene [11] and the IP_3_-induced Ca^2+^ release inhibitor 2-aminoethoxydiphenyl borate (2-APB) [12] reduced the NMDA (6 nmol)-induced dilation of retinal arterioles (Figure 8A), whereas dantrolene, but not 2-APB, prevented NOR3 (5 nmol)-induced retinal vasodilation (Figure 8B).

## 3. Discussion

The present study demonstrates that intravitreal injection of NMDA induces the dilation of retinal arterioles in rats. The NMDA-induced retinal vasodilation was significantly reduced by inhibitors of nNOS and cyclooxygenase, loss of inner retinal neurons, and dysfunction of glial cells. Intravitreal injection of the NO donor, NOR3, also induced the retinal vasodilator response via mechanisms involving glial cells. These results suggest that the activation of NMDA receptors on neuronal cells enhances the nNOS-derived NO production; the released NO induces glial cell-dependent vasodilator response. 

In most vascular beds, NO is known to activate the soluble guanylyl cyclase in vascular smooth muscle cells, leading to increased intracellular cGMP, which, in turn, dilates the blood vessels [13]. Therefore, intravitreally injected NOR3 may induce dilation of retinal blood vessels by activating the NO-induced signaling pathway in vascular cells. However, the results that glial toxins markedly prevented the NOR3-induced retinal vasodilation suggest that the increased NO in the vitreous cavity induces glial cell-dependent vasodilator response.

Stimulation of glial cells results in the release of vasoactive metabolites of arachidonic acid, including the vasodilators PGE_2_ and EETs, which contribute to the neurovascular coupling in the retina [9,10]. However, it has not yet been investigated whether NO stimulates the release of these vasoactive substances from glial cells. In this study, using pharmacological interventions, we found that the intravitreal injection of NOR3 induces retinal vasodilation primarily through the release of vasodilatory prostanoids and EETs from glial cells. In addition to vasodilative substances, glial cells also produce vasoconstrictive metabolites of arachidonic acid such as 20-HETE [9,10]. However, 20-HETE is unlikely to contribute to the NO-induced glial cell-dependent vasodilator response, because 17-octadecynoic acid had no significant effect on the NOR3-induced retinal vasodilation. 

Production of prostanoids and EETs is stimulated by Ca^2+^-dependent activation of phospholipase A_2_. To clarify the mechanism of nNOS-derived NO-mediated stimulation of the Ca^2+^-dependent pathway in glial cells, we examined the possible involvement of ryanodine receptors in the NOR3-induced retinal vasodilation. Ryanodine receptors are responsible for the release of Ca^2+^ from intracellular stores. NO functions as a ryanodine receptor type 1 agonist to release Ca^2+^ from neuronal intracellular Ca^2+^ stores through a reversible S-nitrosylation of the receptor [14]. Interestingly, the ryanodine receptor antagonist dantrolene, but not the IP_3_-induced Ca^2+^ release inhibitor 2-APB, prevented the NOR3-induced retinal vasodilation. These findings suggest that the NO released from neuronal cells could stimulate the production of prostanoids and EETs through the ryanodine receptor-mediated increase in the intracellular Ca^2+^ in glial cells. On the other hand, both dantrolene and 2-APB reduced the NMDA-induced dilation of retinal arterioles. In addition to glial cells, neuronal cells can directly regulate the vascular tone by releasing prostanoids and NO [15]. The additive inhibitory effect of indomethacin and l-NPA on NMDA-induced retinal vasodilation suggests that both nNOS-derived NO and prostanoids are involved in the response. The NO-independent production of vasodilatory prostanoids may be stimulated by an increase in the intracellular Ca^2+^ through the IP_3_-induced Ca^2+^ release mechanism.

Consistent with the previous observations [7,8,16], our immunohistochemical data indicated the presence of nNOS-positive cells with different expression levels in the GCL and INL and nNOS-positive fibers in the IPL. The NMDA receptors have been shown to be expressed in ganglion and amacrine cells in the retina [17,18]. In the present study, we found nNOS-expressing neuronal cells to be highly susceptible to NMDA-induced retinal neurodegeneration. In the neuronal cell loss model, with decreased levels of nNOS proteins, the NMDA-induced retinal vasodilation was markedly diminished. These results suggest that nNOS-positive ganglion and amacrine cells are responsible for the NMDA-induced glial cell-dependent vasodilator response.

The number of nNOS-positive cells in the retina was found to be relatively low. However, a previous study demonstrated that the distribution of nNOS mRNA differs from that of nNOS immunoreactivity in the rat retina [16]. The number of nNOS immunoreactive cells was smaller than the number of cells with intense nNOS mRNA signals [16]. These observations imply that nNOS proteins synthesized in the perikaryons of neuronal cells may be transported into the dendrites and axons. The dense plexus of nNOS-positive processes in the IPL supports this hypothesis. 

In this study, we used three glial toxins (disialoganglioside-GD1b, l-alpha-aminoadipic acid, and fluorocitrate) and found that all the toxins prevented NOR3-induced retinal vasodilation. These glial toxins exert different inhibitory activities on astrocytes and Müller cells [19,20,21,22]. Therefore, there is a possibility that astrocytes, Müller cells, or both contribute to the NMDA-induced retinal vasodilation. Anatomically, astrocytes are limited to the nerve fiber layer, whereas Müller cells run radially from the inner limiting membrane to the outer limiting membrane. The processes of astrocytes wrap around the blood vessels at the retinal surface, whereas the processes and endfeet of Müller cells are in close contact with the blood vessels at the retinal surface and deeper layers. It can be speculated that the nNOS-positive fibers might be closely associated with the processes and endfeet of these glial cells in the IPL. Thus, it is likely that the NO released from neuronal cells could enhance the ryanodine receptor-mediated Ca^2+^ release in glial cells, thereby stimulating the production of prostanoids and EETs in the cells. The released prostanoids and EETs then induce the retinal vasodilator response (Figure 9).

Results from the present study clearly indicate that the NO released from stimulated retinal neuronal cells acts as a key messenger in neuron–glia signaling and contributes to neuronal activity-dependent vasodilation in the retina. However, when retinal NO levels are elevated by the addition of NO donors and induction of type 1 diabetes, glial cell-mediated vasodilation is reduced [2,23]. The flickering light-induced vasodilation of retinal blood vessels is diminished in diabetic patients [24,25]. Therefore, it would be interesting to examine through future studies how the regulatory mechanisms of retinal vascular tone proposed here are affected under diabetic conditions. 

In summary, the present study provides the first pharmacological evidence that stimulation of NMDA receptors on retinal neuronal cells facilitates the nNOS-derived NO production in neuronal cells, which, in turn, induces the vasodilator response primarily through glial-derived EETs and vasodilatory prostanoids. Thus, in response to the neuronal activity, synaptically released glutamate activates the NMDA receptors on retinal neuronal cells, consequently affecting the retinal vascular tone through the neuron–glia–vessel interaction network.

## 4. Materials and Methods

### 4.1. Animals

Wistar rats (male, 7–9 weeks old) were obtained from the Charles River Breeding Laboratories (Tokyo, Japan) and maintained in a room at a constant temperature (22 ± 2 °C), constant humidity (55 ± 5%), and 12 h light/dark cycle. The animals were allowed free access to water and food.

All animal procedures were performed in accordance with the Association for Research in Vision and Ophthalmology Statement for the Use of Animals in Ophthalmic and Vision Research and all animal experimental protocols were approved by the Institutional Animal Care and Use Committee for Kitasato University (approval number: 17-18, approved on 8 December 2017).

### 4.2. In Vivo Experiments

Surgical procedures performed in the present study are described in our previous studies [6,26]. Briefly, rats were anesthetized with pentobarbital sodium (50 mg/kg, intraperitoneal (i.p.); Nacalai Tesque, Kyoto, Japan). Each animal was then placed on a heating pad and a tracheotomy was performed for artificial ventilation. The drug was administered via the catheter placed in the jugular vein, and blood pressure was measured using the femoral artery. To minimize the influence of nerve activity and prevent eye movements, rats were treated with tetrodotoxin (50 µg/kg, intravenous (i.v.); Nacalai Tesque) under artificial ventilation with room air (stroke volume, 10 mL/kg and frequency, 80 strokes/min) using a rodent respirator (SN-480-7; Shinano, Tokyo, Japan). Tetrodotoxin decreased the systemic blood pressure; therefore, methoxamine hydrochloride (30–60 µg/kg/min; Sigma-Aldrich, St. Louis, MO, USA) was continuously injected into the jugular vein at a constant rate using a syringe pump (Model 1140-001; Harvard Apparatus, Holliston, MA, USA) to maintain an adequate systemic circulation.

#### 4.2.1. Effects of Intravitreal Injection of NMDA on the Diameter of Retinal Arterioles, Blood Pressure, and Heart Rate (Protocol 1)

We first examined the effects of intravitreal injection of NMDA on the diameter of retinal arterioles, blood pressure, and heart rate. NMDA (6 nmol; Nacalai Tesque) was injected into the vitreous cavity of one eye of the rats. The same volume of the vehicle (saline) was injected into the vitreous cavity of one eye of the animals that served as a control. We found that NMDA at a dose of 6 nmol induced approximately 80% maximum response in preliminary studies.

#### 4.2.2. Role of NO and Prostaglandins in Retinal Vasodilator Responses to NMDA (Protocol 2)

To examine the possible involvement of NO and vasodilatory prostanoids in NMDA-induced vasodilation in the retina, l-NAME (a non-selective NOS inhibitor, 40 nmol; Sigma-Aldrich), l-NPA (a selective nNOS inhibitor, 400 nmol; Cayman Chemical Co., Ann Arbor, MI, USA), indomethacin (a cyclooxygenase inhibitor, 10 nmol; Sigma-Aldrich), l-NPA (400 nmol) + indomethacin (10 nmol), or vehicle for each inhibitor was intravitreally administered before the surgical procedures and tetrodotoxin treatment. Doses of the inhibitors were selected based on our previous study [26]. We had previously found that the dose of l-NPA did not affect the retinal vasodilator responses induced by intravenous infusion of acetylcholine (1–10 µg/kg/min), an endothelium-dependent vasodilator [26]. Therefore, l-NPA did not seem to have any significant effect on the endothelial NOS under our experimental conditions.

#### 4.2.3. Role of Neuronal Cells in Retinal Vasodilator Responses to NMDA (Protocol 3)

A marked loss of inner retinal neurons including ganglion and amacrine cells was observed 7 days following the intravitreal injection of NMDA (200 nmol) [26]. Therefore, we named this excitotoxicity model the “neuronal cell loss model.” The NMDA (6 nmol)-induced vasodilator response was assessed in the neuronal cell loss model. The same volume of the vehicle (saline) injected into the vitreous cavity of one eye of the animals that served as the control.

#### 4.2.4. Role of Glial Cells in Retinal Vasodilator Responses to NMDA and NOR3 (Protocols 4 and 5)

We examined the effects of intravitreal injection of glial toxin disialoganglioside-GD1b (15 nmol; Sigma-Aldrich) on NMDA (6 nmol)-induced increase in the diameter of retinal arterioles (Protocol 4). In another set of experiments (Protocol 5), the effect of intravitreal injection of (±)-(E)-4-ethyl-2-[(E)-hydroxyimino]-5-nitro-3-hexenamide (NOR3, 5 nmol; Dojindo, Kumamoto, Japan) on the diameter of retinal arterioles and effects of glial toxins (disialoganglioside-GD1b (15 nmol), l-alpha-aminoadipic acid (250 nmol; Sigma-Aldrich), and dl-fluorocitric acid barium salt (fluorocitrate, 450 pmol; Sigma-Aldrich) on the NOR3-induced response was studied. A total volume of 3 µL of each glial toxin was injected into the vitreous cavity of one eye before the surgical procedures and tetrodotoxin treatment as described above. The same volume of the vehicle for each glial toxin was injected into the vitreous cavity of one eye of animals that served as a control. In a previous study, we found that retinal vasodilation induced by intravenous infusion of NOR3 (1–10 µg/kg/min) was unaffected by the intravitreal injection of glial toxins [6]. Therefore, glial toxins seemed to have no significant effect on the responsiveness of retinal arterioles to NO under our experimental conditions.

#### 4.2.5. Involvement of Prostaglandins, EETs, and 20-HETE in Retinal Vasodilator Responses to NO (Protocol 6)

To identify the factors released from NO-stimulated glial cells, we examined the effects of intravitreal injection of 14,15-EE-5(Z)-E (an analog of EET, 0.7 nmol; Cayman Chemical), 17-octadecynoic acid (an inhibitor of ω-hydroxylase, which mediates conversion of arachidonic acid to 20-HETE, 1.4 nmol; Cayman Chemical), and indomethacin (10 nmol) on intravitreal injection of NOR3 (5 nmol)-induced dilation of retinal arterioles.

#### 4.2.6. Mechanisms of Release of Glial Cell-Derived Factors (Protocol 7)

To determine the regulatory mechanisms mediating the release of glial cell-derived factors, we examined the effects of intravitreal injection of dantrolene (1.7 nmol; Sigma-Aldrich), a ryanodine receptor antagonist [11], and 2-APB (6.4 nmol; Sigma-Aldrich), a membrane-permeable blocker of the IP_3_-induced Ca^2+^ release [12], on intravitreal injection of NMDA (6 nmol)- and NOR3 (5 nmol)-induced dilation of retinal arterioles.

### 4.3. Fundus Photography and Retinal Arteriolar Diameter Measurement

The fundus images were captured using a digital camera (EOS7D; Canon, Tokyo, Japan) equipped with a bore scope-type objective lens for small animals (Model 01; Scalar, Tokyo, Japan) as described in our previous studies [6,26]. A region (138 × 276 µm) of the fundus image (3456 × 5184 µm) containing a retinal arteriole was selected for analysis. The diameter of the vessel in the same region was measured throughout the experiment. 

### 4.4. Immunohistochemistry

To determine the co-localization of nNOS with neuronal and glial cells in the rat retina, immunohistochemical staining of whole-mounts and cross-sections was performed in a separate group of rats, as previously described [6]. 

Primary antibodies used were rabbit monoclonal anti-nNOS antibody (1:500, clone C7D7; Cell Signaling Technology, Danvers, MA, USA), mouse monoclonal anti-βIII tubulin antibody (1:1000; Promega, Madison, WI, USA), Alexa Fluor 488-conjugated mouse monoclonal anti-GFAP antibody (1:2000, clone G-A-5; Thermo Fisher Scientific, Waltham, MA, USA), mouse monoclonal anti-calretinin antibody (1:2000; Millipore, Temecula, CA, USA), goat polyclonal anti-glycine transporter 1 antibody (1:500; Millipore), rabbit polyclonal anti-tyrosine hydroxylase antibody (1:5000; Millipore), mouse monoclonal anti-parvalbumin antibody (1:500; Sigma-Aldrich), and mouse monoclonal anti-calbindin-D-28K antibody (1:2000; Sigma-Aldrich). The secondary antibodies used were fluorescence (Alexa Fluor 488 or Cy3)-conjugated, species-specific secondary antibodies (1:400; Jackson ImmunoResearch Laboratories, West Grove, PA, USA).

Images were obtained with the fluorescent microscope system (BZ-9000; Keyence, Osaka, Japan) or the confocal laser scanning microscope (LSM 710; Zeiss, Oberkochen, Germany). The number of nNOS-positive cells present in the GCL and INL, and the nNOS-positive area in the IPL of each retina were determined.

### 4.5. Data Analysis

The retinal arteriolar diameter, mean arterial pressure, and heart rate were expressed as a percentage (%) of the baseline level (mean values of the data obtained from time −2 to 0 min). In Protocols 2, 6, and 7, changes in the diameters of retinal arterioles following the intravitreal injection of NMDA and NOR3 were assessed by measuring the mean diameters from time 20 to 30 min and time 10 to 15 min, respectively. All values are presented as the mean ± standard error (SE). An unpaired *t*-test and one-way analysis of variance (ANOVA), followed by the Tukey post-test, were used to compare baseline values between the two groups and among more than two groups, respectively. When comparing the responses to NMDA or NOR3, a two-way ANOVA was used (PRISM 6; GraphPad Software, San Diego, CA, USA). A *p*-value less than 0.05 was considered to be statistically significant.

## Figures and Tables

**Figure 1 ijms-20-01952-f001:**
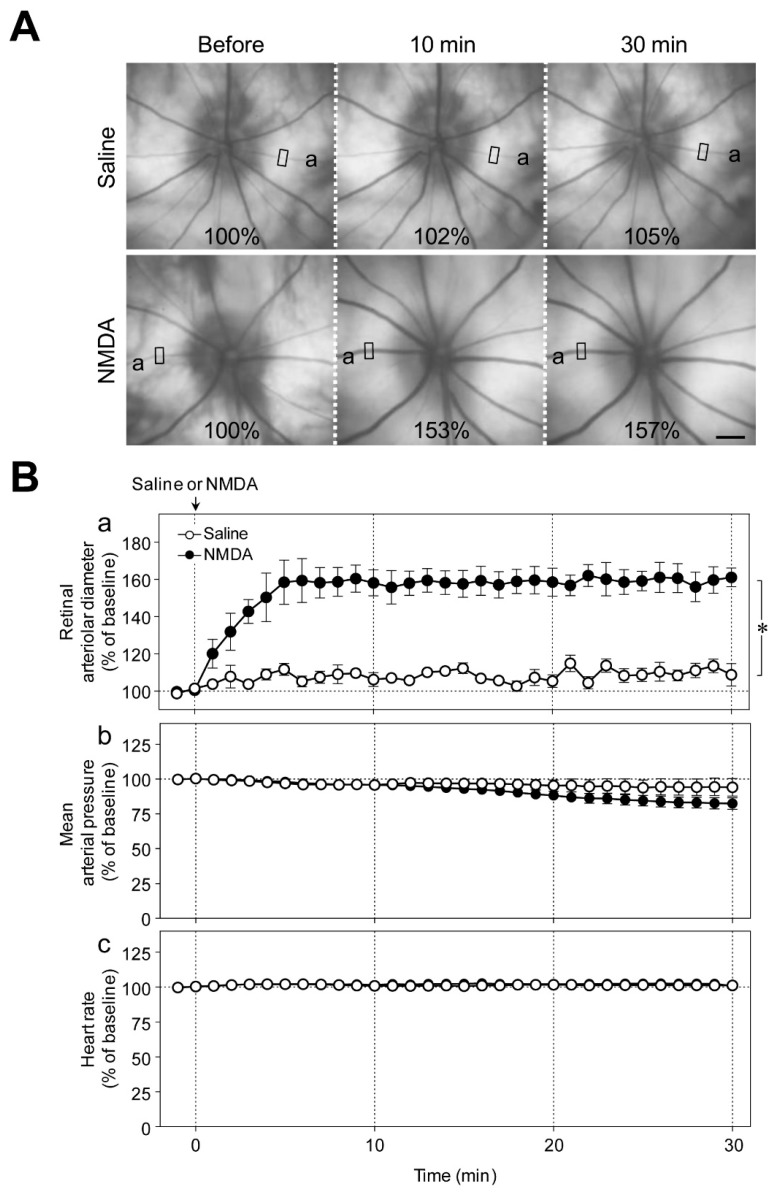
Effects of intravitreal injection of NMDA on retinal blood vessels in rats. (**A**) Representative fundus images before, and 10 and 30 min after intravitreal injection of saline and NMDA (6 nmol). Values indicate the diameter of the retinal arteriole in the selected region expressed as a percentage of the baseline value. Scale bar: 500 μm. (**B**) Changes in (**a**) the retinal arteriolar diameter, (**b**) mean arterial pressure, and (**c**) heart rate induced by intravitreal injection of saline or NMDA. Each point with a vertical bar represents the mean ± SE from five animals. * *p* < 0.05.

**Figure 2 ijms-20-01952-f002:**
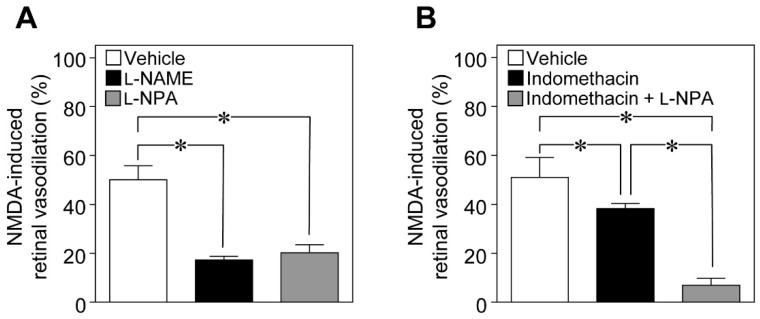
Involvement of nitric oxide (NO) and prostaglandins in the retinal vasodilator response induced by intravitreal injection of NMDA. (**A**) NMDA-induced changes in the retinal arteriolar diameter in the presence of *N*^ω^-nitro-l-arginine methyl ester (l-NAME) (40 nmol), *N*^ω^-propyl-l-arginine (l-NPA) (400 nmol), or the vehicle. (**B**) NMDA (6 nmol)-induced changes in the retinal arteriolar diameter in the presence of indomethacin (10 nmol), indomethacin + l-NPA (400 nmol), or the vehicle. Each column with a vertical bar represents the mean ± SE from four to six animals. * *p* < 0.05.

**Figure 3 ijms-20-01952-f003:**
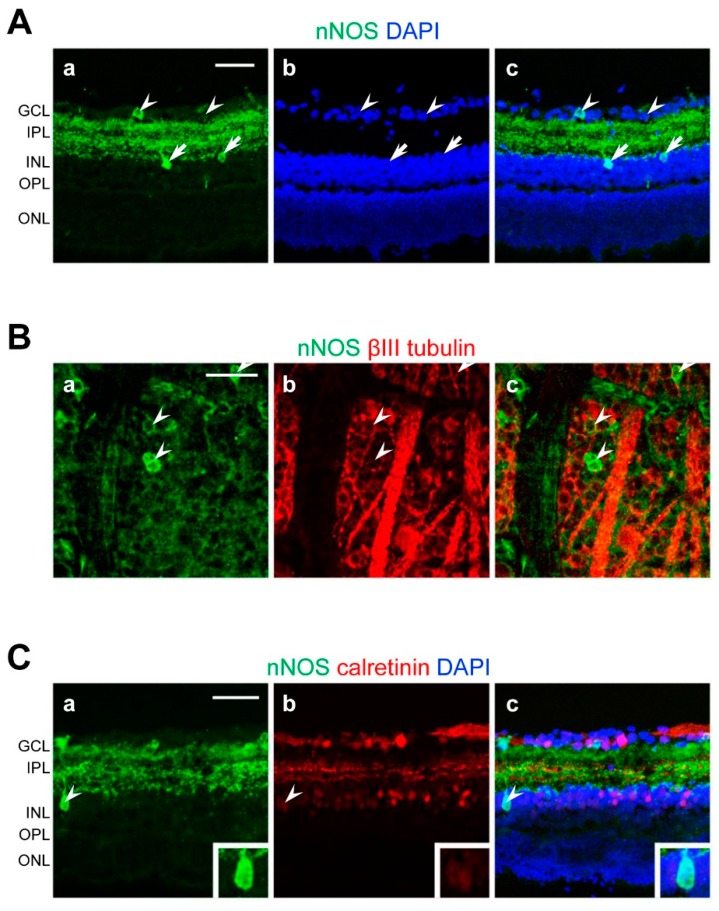
Localization of immunoreactivities of neuronal NO synthase (nNOS) in the retina. Confocal microscopy images of retinal flat-mounts and cross-sections labeled with anti-nNOS, anti-βIII-tubulin, or anti-calretinin antibodies. (**A**) Distribution of nNOS immunoreactivities. (**a**–**c**) Arrowheads and arrows indicate nNOS-positive cells in the ganglion cell layer (GCL) and inner plexiform layer (INL), respectively. (**B**) Co-localization of nNOS immunoreactivities with βIII-tubulin-positive ganglion cells. (**a**–**c**) Arrowheads indicate nNOS- and βIII-tubulin-double positive cells. (**C**) Co-localization of nNOS immunoreactivities with calretinin-positive amacrine cells. (**a**–**c**) Arrowheads indicate calretinin-positive amacrine cells with strong nNOS immunoreactivity. A higher magnification image of the cell is shown in the inset. Scale bars: (**A**–**C**) 50 μm in **a** (applies to **b**,**c**). DAPI, 4′,6-diamidino-2-phenylindole; IPL, inner plexiform layer; OPL, outer plexiform layer; ONL, outer nuclear layer.

**Figure 4 ijms-20-01952-f004:**
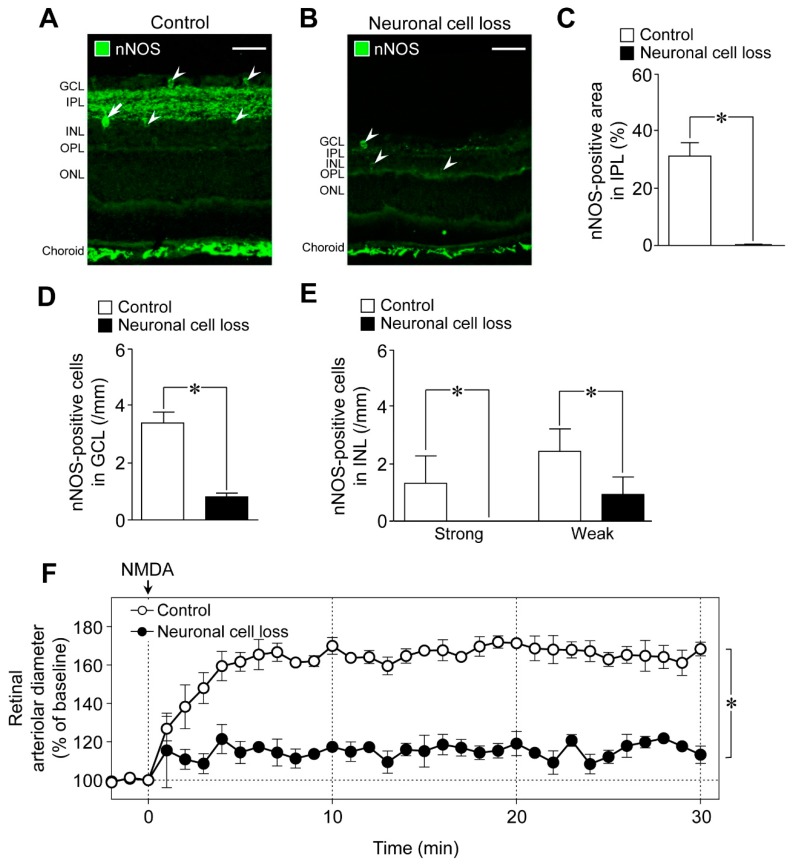
Effects of loss of inner retinal neurons on the number of neuronal NO synthase (nNOS)-positive cells and the retinal vasodilation induced by intravitreal injection of NMDA. (**A**,**B**) Confocal microscopy images of retinal cross-sections labeled with (**A**) anti-nNOS antibodies from control (**B**) and neuronal cell loss model. Arrows and arrowheads indicate nNOS-positive cells with strong and weak immunoreactivities, respectively. Scale bars: (**A**,**B**) 50 μm. (**C**–**E**): Quantification of (**C**) nNOS-positive area (**D**,**E**) and cell number. Each column with a vertical bar represents the mean ± SE from five animals. * *p* < 0.05. (**F**) Changes in the retinal arteriolar diameter induced by intravitreal injection of NMDA (6 nmol) in control and neuronal cell loss model. Each point with a vertical bar represents the mean ± SE from four animals. * *p* < 0.05. GCL, ganglion cell layer; IPL, inner plexiform layer; INL, inner nuclear layer; OPL, outer plexiform layer; ONL, outer nuclear layer.

**Figure 5 ijms-20-01952-f005:**
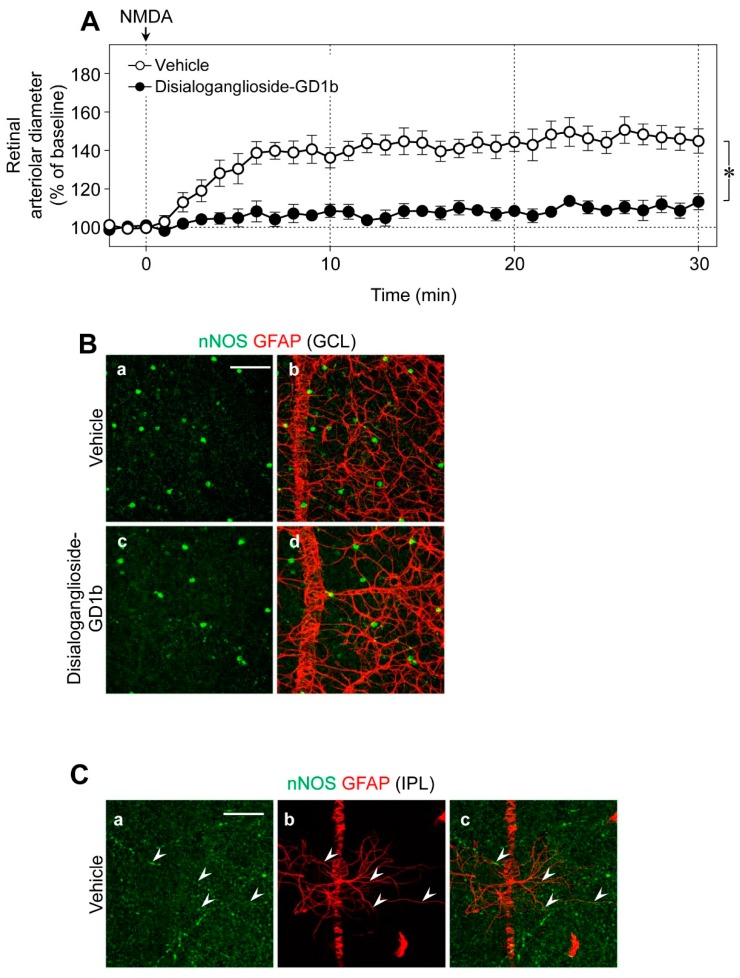
Effects of disialoganglioside-GD1b on the retinal vasodilation induced by intravitreal injection of NMDA and nNOS- and glial fibrillary acidic protein (GFAP)-positive positive cells. (**A**) NMDA (6 nmol)-induced changes in the retinal arteriolar diameter in the presence of disialoganglioside-GD1b (15 nmol) or the vehicle. Each point with a vertical bar represents the mean ± SE from five to seven animals. * *p* < 0.05. (**B**,**C**) Confocal microscopy images of retinal flat-mounts labeled with anti-nNOS and anti-GFAP antibodies. In panel (**C**), arrowheads indicate that nNOS-positive fibers were closely associated with the processes of GFAP-positive astrocytes. Scale bars: (**B**) 100 μm in **a** (applies to **b**–**d**) and (**C**) 50 μm in **a** (applies to **b**,**c**). GCL, ganglion cell layer; IPL, inner plexiform layer.

**Figure 6 ijms-20-01952-f006:**
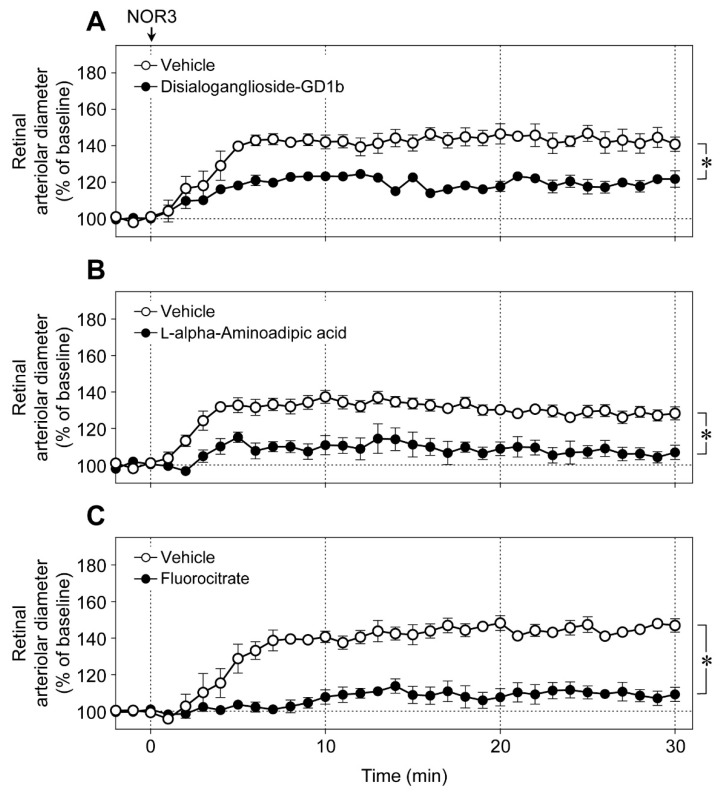
Effects of glial toxins on the retinal vasodilation induced by intravitreal injection of NOR3. NOR3 (5 nmol)-induced changes in retinal arteriolar diameter in (**A**) the presence of disialoganglioside-GD1b (15 nmol), (**B**) l-alpha-aminoadipic acid (250 nmol), (**C**) fluorocitrate (450 pmol) or the vehicle. Each point with a vertical bar represents the mean ± SE from three to seven animals. * *p* < 0.05.

**Figure 7 ijms-20-01952-f007:**
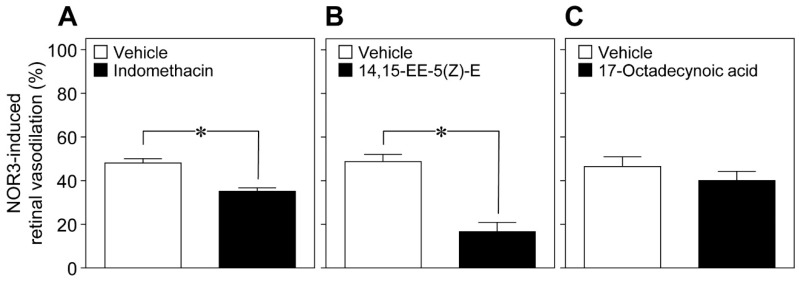
Involvement of prostaglandins and epoxyeicosatrienoic acids in the retinal vasodilation induced by intravitreal injection of NOR3. NOR3 (5 nmol)-induced changes in the retinal arteriolar diameter in the presence of (**A**) indomethacin (10 nmol), (**B**) 14,15-EE-5(Z)-E (0.7 nmol), (**C**) and 17-octadecynoic acid (1.4 nmol). Each column with a vertical bar represents the mean ± SE from four animals. * *p* < 0.05.

**Figure 8 ijms-20-01952-f008:**
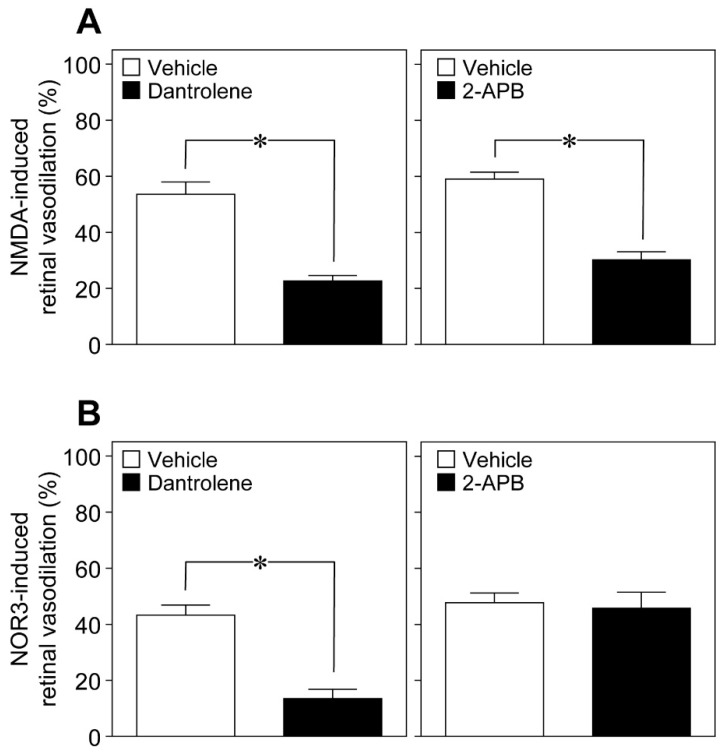
Involvement of ryanodine receptors and inositol 1,4,5-trisphosphate (IP_3_) receptors in the retinal vasodilation induced by intravitreal injection of NMDA and NOR3. (**A**) NMDA (6 nmol)-induced changes in the retinal arteriolar diameter in the presence of dantrolene (1.7 nmol) and 2-aminoethoxydiphenyl borate (2-APB) (6.4 nmol). (**B**) NOR3 (5 nmol)-induced changes in the retinal arteriolar diameter in the presence of dantrolene (1.7 nmol) and 2-APB (6.4 nmol). Each column with a vertical bar represents the mean ± SE from four animals. * *p* < 0.05.

**Figure 9 ijms-20-01952-f009:**
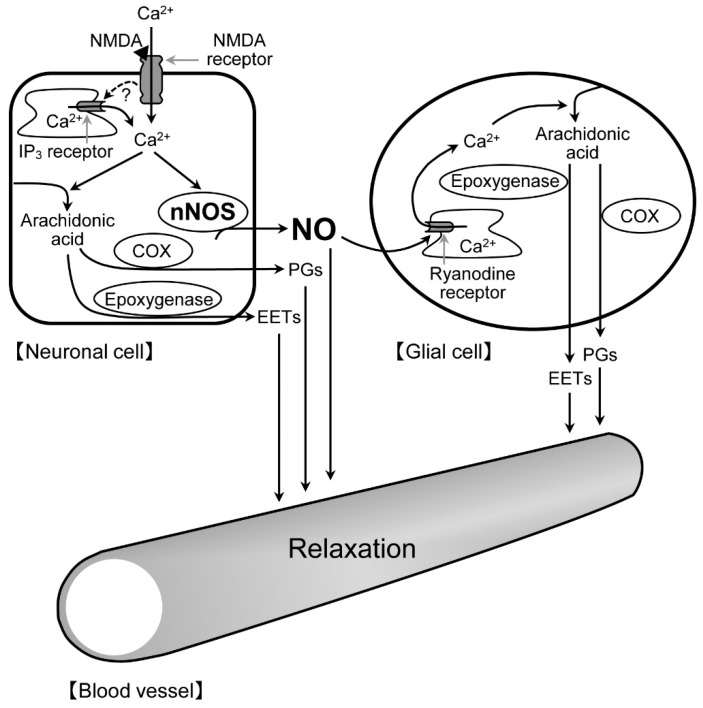
A putative mechanistic model for the interaction between neuronal cells and glial cells in the retinal vasodilatory mechanism. Stimulation of NMDA receptors on neuronal cells facilitates neuronal nNOS-derived NO production. The NO released from neuronal cells acts on glial cells and enhances ryanodine receptor-mediated Ca^2+^ release and thereby stimulates the production of PGs and EETs in the cells. The released PGs and EETs dilate retinal blood vessels. A black triangle mark (▲) indicates NMDA. EETs, epoxyeicosatrienoic acids; NMDA, *N*-methyl-d-aspartic acid; NO, nitric oxide; nNOS, neuronal NO synthase; PGs, prostaglandins.

**Table 1 ijms-20-01952-t001:** Baseline values of retinal arteriolar diameter (AD), mean arterial pressure (MAP), and heart rate (HR) before intravitreal injection of saline or *N*-methyl-d-aspartic acid (NMDA).

	AD (µm)	MAP (mmHg)	HR (beats/min)
Protocol 1			
Saline (*n* = 5)	42.0 ± 0.9	115 ± 2	337 ± 21
NMDA (*n* = 5)	37.3 ± 0.8	119 ± 2	332 ± 10

**Table 2 ijms-20-01952-t002:** Baseline values of retinal arteriolar diameter in each experimental group.

	AD (µm)
Protocol 2	
Vehicle (*n* = 4)	45.0 ± 2.7
*N*^G^-nitro-l-arginine methyl ester (*n* = 4)	47.7 ± 1.9
*N*^G^-propyl-l-arginine (*n* = 4)	50.4 ± 3.0
Vehicle (*n* = 4)	37.3 ± 2.5
Indomethacin (*n* = 4)	38.0 ± 3.8
Indomethacin+ *N*^G^-propyl-l-arginine (*n* = 6)	37.3 ± 2.2
Protocol 3	
Control (*n* = 4)	37.7 ± 2.0
Neuronal cells loss (n=4)	38.4 ± 3.7
Protocol 4	
Vehicle (*n* = 7)	43.6 ± 1.6
Disialoganglioside-GD1b (*n* = 5)	42.6 ± 2.2
Protocol 5	
Vehicle (*n* = 4)	41.7 ± 1.0
Disialoganglioside-GD1b (*n* = 3)	38.5 ± 4.1
Vehicle (*n* = 7)	43.2 ± 1.4
l-alpha-Aminoadipic acid (*n* = 5)	44.1 ± 1.9
Vehicle (*n* = 4)	43.5 ± 1.7
Fluorocitrate (*n* = 4)	44.4 ± 3.6
Protocol 6	
Vehicle (*n* = 4)	37.2 ± 0.5
Indomethacin (*n* = 4)	39.1 ± 1.1
Vehicle (*n* = 4)	40.8 ± 1.8
14,15-EE-5(Z)-E (*n* = 4)	44.9 ± 1.4
Vehicle (*n* = 4)	36.3 ± 0.7
17-octadecynoic acid (*n* = 4)	39.5 ± 1.4
Protocol 7	
Vehicle (*n* = 8)	40.5 ± 1.8
Dantrolene (*n* = 8)	40.5 ± 1.4
Vehicle (*n* = 8)	37.2 ± 1.8
2-aminoethoxydiphenyl borate (*n* = 8)	40.7 ± 1.3

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
