# Peer review of "Role of Neuron–Glia Signaling in Regulation of Retinal Vascular Tone in Rats"

_ijms, 2019, doi:10.3390/ijms20081952_

Reviewer 1 Report

In the current study, Someya et al have presented information that expands on their previously detailed mechanism of retinal vascular regulation with particular focus on the role of NMDA induced nNOS production and vasodilation.

This paper eloquently builds on the authors previous findings, is well presented and is concluded with a detailed scheme that brings the significant results together.

I have only one minor comment that the authors can choose to address.

The clinical significance of these findings are touched on only briefly at the very end of the paper where the authors suggest it would be interesting to study this mechanism in diabetic conditions. Further exploration of this thought would enhance the findings of the study and appeal to a wider audience.

Author Response

Reviewer 1

Comments and Suggestions for Authors

In the current study, Someya et al have presented information that expands on their previously detailed mechanism of retinal vascular regulation with particular focus on the role of NMDA induced nNOS production and vasodilation.

This paper eloquently builds on the authors previous findings, is well presented and is concluded with a detailed scheme that brings the significant results together.

I have only one minor comment that the authors can choose to address.

The clinical significance of these findings are touched on only briefly at the very end of the paper where the authors suggest it would be interesting to study this mechanism in diabetic conditions. Further exploration of this thought would enhance the findings of the study and appeal to a wider audience.

[Response] We thank the reviewer for the positive and insightful comment on our manuscript. As suggested by the reviewer, we revised the part of the discussion in the revised manuscript (Page 12, Line3-8).

Reviewer 2 Report

In this paper Someya et al. explain how intravitreal injections of NDMA lead to vascular dilation.

The authors first show that vascular dilation was not associated to an increased arterial pressure neither to the increase of the heart rate (Fig.1)

Using selective (L-NPA) and non-selective (L-NAME) NO inhibitors and the prostaglandin synthesis inhibitor Indomethacin, they then show that the observed vasodilation was due to NO and prostaglandins (Fig.2)

Immunostaining panels show the presence of nNOS in ganglion cells (bIII tubulin) and amacrine cells (calretinin+), while it is not present in glial cells (Fig.3, Supp.1)

Using a “neuronal loss model” they show that prolonged treatment with NMDA preferentially disrupts nNOS-expressing neurons and prevented NMDA-mediated vasodilation (Fig.4).

The authors previously showed that in the neuronal loss model the responsiveness of retinal arterioles to NO remained unaltered. Therefore they suggest that glial cells are the real responsible for the vascular response to NMDA.

Using glial toxins they show that glial dysfunction markedly attenuated NMDA-mediated vasodilation (Fig.5).

They also show that glial toxins prevented NOR3-mediated vasodilation, excluding neuronal-dependent vasodilation response (Fig. 6).

The authors show using pharmacological inhibition that NO stimulates the glial cells to preferentially release vasodilator metabolites (Fig.7).

Finally the authors show the involvement of ryanodine receptors and IP3 receptors in the NMDA- and NOR3-mediated retinal vasodilation (Fig.8)

The manuscript is concisely written, clear and the figures support the conclusions.

Concerns:

1) It is known that glial cells control vessel response through release of vasodilator molecules such as prostaglandins and EETs or vasoconstrictor molecules such as 20-HETE (see Newman, 2015). The authors should better empathize the novelty of their study.

2) The authors should better explain early in the text what molecules like NOR3, Indomethacin are, the reader has to get to the discussion to gather the information.

3) In this study the authors used 3 different glial toxins (disialoganglioside-GD1b, L-alpha-aminoadipic acid and fluorocitrate) to induce glial dysfunction, however they don’t explain in the text the specific effect that each of these toxins have on the glial cells population and their possible side effects on endothelial cells’ function.

4) When treating rats with glial toxins, the authors show that there is not defect in the neuronal population (GFAP+ and nNOS+). However, the authors need to show the morphology of the retinal vasculature after treatment with glial toxins to exclude any vascular defect when glial cells are not properly functioning

5) Please specify the gender of rats in the material and methods section.

Author Response

Reviewer 2

Comments and Suggestions for Authors

In this paper Someya et al. explain how intravitreal injections of NDMA lead to vascular dilation.

The authors first show that vascular dilation was not associated to an increased arterial pressure neither to the increase of the heart rate (Fig.1)

Using selective (L-NPA) and non-selective (L-NAME) NO inhibitors and the prostaglandin synthesis inhibitor Indomethacin, they then show that the observed vasodilation was due to NO and prostaglandins (Fig.2)

Immunostaining panels show the presence of nNOS in ganglion cells (bIII tubulin) and amacrine cells (calretinin+), while it is not present in glial cells (Fig.3, Supp.1)

Using a “neuronal loss model” they show that prolonged treatment with NMDA preferentially disrupts nNOS-expressing neurons and prevented NMDA-mediated vasodilation (Fig.4).

The authors previously showed that in the neuronal loss model the responsiveness of retinal arterioles to NO remained unaltered. Therefore they suggest that glial cells are the real responsible for the vascular response to NMDA.

Using glial toxins they show that glial dysfunction markedly attenuated NMDA-mediated vasodilation (Fig.5).

They also show that glial toxins prevented NOR3-mediated vasodilation, excluding neuronal-dependent vasodilation response (Fig. 6).

The authors show using pharmacological inhibition that NO stimulates the glial cells to preferentially release vasodilator metabolites (Fig.7).

Finally the authors show the involvement of ryanodine receptors and IP3 receptors in the NMDA- and NOR3-mediated retinal vasodilation (Fig.8)

The manuscript is concisely written, clear and the figures support the conclusions.

[Response] We thank the reviewer for the positive and insightful comment on our manuscript. We have addressed each of these and made corresponding changes in the manuscript.

Concerns:

1) It is known that glial cells control vessel response through release of vasodilator molecules such as prostaglandins and EETs or vasoconstrictor molecules such as 20-HETE (see Newman, 2015). The authors should better empathize the novelty of their study.

[Response] As pointed out by the reviewer, Newman et al. demonstrated that glial cells control vessel response through release of vasodilator molecules such as prostaglandins and EETs or vasoconstrictor molecules such as 20-HETE. However, they did not address the question of whether NO stimulates the release of these vasoactive substances from glial cells. Therefore, we added one sentence “However, it has not yet been investigated whether NO stimulates the release of these vasoactive substances from glial cells (Page 10, Line23-24).”

2) The authors should better explain early in the text what molecules like NOR3, Indomethacin are, the reader has to get to the discussion to gather the information.

[Response] As suggested by the reviewer, we revised the manuscript.

3) In this study the authors used 3 different glial toxins (disialoganglioside-GD1b, L-alpha-aminoadipic acid and fluorocitrate) to induce glial dysfunction, however they don’t explain in the text the specific effect that each of these toxins have on the glial cells population and their possible side effects on endothelial cells’ function.

[Response] This is an important point. As pointed out by the reviewer, glial toxins (disialoganglioside-GD1b, L-alpha-aminoadipic acid and fluorocitrate) used in this study exert different inhibitory activities on astrocytes and MĂĽller cells. Nevertheless, the present results indicate that these toxins show the similar preventive effects on NOR3-induced retinal vasodilation. Therefore, it is unclear whether astrocytes, MĂĽller cells, or both contribute to the NMDA-induced retinal vasodilation. To address this point, we added two sentences as follows.

[Page 11, Line 9-11] In this study, we used three glial toxins (disialoganglioside-GD1b, L-alpha-aminoadipic acid and fluorocitrate) and found that all toxins prevented NOR3-induced retinal vasodilation. These glial toxins exert different inhibitory activities on astrocytes and MĂĽller cells [20-23].

We cannot exclude the possibility that glial toxins have any side effects on endothelial cells’ function. However, we found that retinal vasodilation induced by intravenous infusion of NOR3 are unaffected by intravitreal injection of these three glial toxins. Therefore, we think that the responsiveness of retinal arterioles to NO is unaffected by treatment with glial toxins.

4) When treating rats with glial toxins, the authors show that there is not defect in the neuronal population (GFAP+ and nNOS+). However, the authors need to show the morphology of the retinal vasculature after treatment with glial toxins to exclude any vascular defect when glial cells are not properly functioning.

[Response] Unfortunately, we did not examine the morphology of the retinal vasculature after treatment with glial toxins. However, as described above, we found that the responsiveness of retinal arterioles to NO is unaffected by these glial toxins.

5) Please specify the gender of rats in the material and methods section.

[Response] We put the gender information.